# Modelling the factors affecting the probability for local rabies elimination by strategic control

**Johann L. Kotzé** [1]*, **John Duncan Grewar** [1,2], **Aaron Anderson** [3]

**1** Department of Production Animal Studies, University of Pretoria, Onderstepoort, South Africa, **2** jDATA (Pty) Ltd, Sandbaai, South Africa, **3** USDA National Wildlife Research Centre, Fort Collins, Colorado, United States of America

* johann.vet@gmail.com

## Abstract

Dog rabies has been recognized from ancient times and remains widespread across the developing world with an estimated 59,000 people dying annually from the disease. In 2011 a tri-partite alliance consisting of the OIE, the WHO and the FAO committed to globally eliminating dog-mediated human rabies by 2030. Regardless of global support, the responsibility remains with local program managers to implement successful elimination programs. It is well known that vaccination programs have a high probability of successful elimination if they achieve a population-coverage of 70%. It is often quoted that reducing population turnover (typically through sterilizations) raises the probability for local elimination by maintaining herd immunity for longer. Besides this, other factors that affect rabies elimination are rarely mentioned. This paper investigates the probability for local elimination as it relates to immunity, fecundity, dog population size, infectivity (bite rates), in-migration of immune-naïve dogs, and the initial incidence. To achieve this, an individual-based, stochastic, transmission model was manipulated to create a dataset covering combinations of factors that may affect elimination. The results thereof were analysed using a logistic regression model with elimination as the dependent variable. Our results suggest that smaller dog populations, lower infectivity and lower incidence (such as when epidemics start with single introductions) strongly increased the probability for elimination at wide ranges of vaccination levels. Lower fecundity and lower in-migration had weak effects. We discuss the importance of these findings in terms of their impact and their practical application in the design of dog-mediated rabies control programs.

## Author summary

Most guidelines for rabies control call for at least 70% vaccination coverage of dogs. This level of immunity has a very high probability for the local elimination of rabies, but it is often not an achievable ideal due to resource constraints. Campaign managers can be strategic on how they allocate their resources. Lower infectivity rates are present in areas with more restricted dog movements and have higher probabilities for elimination at lower vaccination rates. Smaller sub-populations have higher probabilities for elimination at the

**Data Availability Statement:** Supporting information and additional intermediate data can be sourced at https://doi.org/10.5281/zenodo.4362497.

**Funding:** The author(s) received no specific funding for this work.

**Competing interests:** The authors have declared that no competing interests exist.

same vaccination coverage levels compared to larger sub-populations. Vaccinating immune corridors can divide meta-populations into smaller sub-populations that are likely to result in elimination either due to their small size or due to the local low infectivity. Areas already free of rabies require lower vaccination levels to maintain freedom compared to endemic areas. Where donors do not specifically require sterilization campaigns, funds meant for rabies control should not be diverted to sterilizations.

## Introduction

Dogs are responsible for up to 99% of all human deaths attributed to rabies, transmitting the virus primarily through bites [1]. Exposure to virus does not invariably lead to disease, but when it does the case fatality rate approaches 100% [2]. Rabies has persisted since ancient times, being the oldest known infectious disease to man [3]. Whereas notable successes have been achieved to control dog-transmitted rabies in many countries, most of the globe is still in the endemic state. Despite its widespread prevalence, rabies remains rare (relative to other infectious diseases) with typical annual incidence rates in endemic countries below 0.2% [4]. This is a result of basic reproductive numbers mostly below 1.5 [5].

Good quality rabies vaccines are available, leaving the only remaining barrier to exit the endemic state the ability to get vaccine into dogs and achieve adequate herd immunity [6,7]. It is widely quoted that campaign managers should aim to achieve 70% vaccination coverage [8–10]. However, calculating the required coverage through the basic reproductive number, $R_0$ [11] gives much lower required coverages. Typically, these are between 20% and 40%. Factoring in the waning of vaccine induced immunity and the loss of immune individuals through rapid population turnover rates, the required annual coverage again approaches 70% [4,12].

Despite the common recommendation of 70% coverage, the actual required coverage will vary considerably between locations as a function of the local dog ecology [5,13,14]. Furthermore, budgetary resources at the local level are often not enough to achieve a 70% coverage [7]. Since most of the costs of vaccination campaigns are logistic in nature, achieving a uniform coverage across large meta-populations carries the greatest cost. Instead, campaign managers typically achieve patchy, irregular coverage based on convenience or as a response to reported cases. The ability to strategically select specific, spatially separated sub-populations for campaigns could achieve better long-term results but campaign managers are currently not equipped with enough knowledge to make evidence-based choices. There is therefore a need for a better quantitative understanding of the impact various factors have on the required vaccination coverage at the sub-population level. There is also limited literature on the impact of factors other than vaccination, dog densities and fecundity [15].

Field experiments would be ideal to answer these questions but conducting them may be unethical or impractical. This study attempts to address these needs through an individual-based, bioeconomic, stochastic disease model. Bio-economic models allow for flexibility in both disease parameters as well as management interventions, thereby allowing for their optimization [16]. Stochastic methods allow parameters to take on a constantly changing value based on a probability distribution rather than a fixed value. This is important for diseases with high demographic variability such as rabies [17]. For example, the number of secondary contacts from a rabid dog approaches a negative binomial distribution with mean 2.2 and variance 5.6 [4]. Ignoring such frequency distributions in a mathematical model can have substantial impacts on the outcome [17].

Our hope with this paper is to assist campaign managers to allocate their resources strategically to geographically separate sub-populations based on their relative importance to the sustainability of rabies in the larger meta-population.

## Methods

Seven parameters of a published rabies transmission model were selected to be investigated for their impact on the probability for dog rabies elimination [16] (Table 1). The selected parameters represent factors that are either known or suspected to affect the local elimination of rabies. A range of possible values were chosen for each parameter reflecting the most likely field values. The specified values of the seven parameters created 103,125 unique combinations. Due to the model's stochastic nature, we ran five iterations for each combination. The results of the transmission model were captured in a dataset containing 425,802 successful eliminations and 88,940 sustained outbreaks, each linked to a specific set of values for the seven parameters. We excluded from the analysis 883 iterations where the dog population became extinct. The values of the different parameters that had extinction events are presented in S1 Table and the calculations are shown in S1 Script.

**Table 1. Model parameters included in the transmission model.**

| Model parameter[a] | Parameter space | Comments |
|---|---|---|
| **population size** | {200; 600; 1,800; 5,400; 16,200} Set = 5 | The initial population size at the start of the iteration. The parameter progressively increases from 200 to 16200 by a factor of 3. The default value for calculations was chosen as 5,400 (closest to the mean of 4,840). |
| **infectivity** | {$0.5 \leq x \leq 1.5$} Set = 5 | A factor that is multiplied with the rate of effective bites (0.02252 bites per rabid animal per day). An effective bite is infective and implies infectivity. The default value is 1. |
| **fecundity** | {$0.5 \leq x \leq 1.5$} Set = 5 | A factor that is multiplied with the default mean litter size (4.4) and the default mean litters per year for each fertile female (0.31). The default value is 1. |
| **in-migration** | {$0 \leq x \leq 0.1$} Set = 5 | The in-migration allowed per annum as a fraction of the current population size. The chosen values are lower than the published value of 0.28. This is because high in-migration rates would substantially reduce the ability to observe the effects of the other parameters. In-migrants are assumed immune naïve and not infective. The default value was chosen as 0. |
| **exposure survival** | {$0 \leq x \leq 0.001$} Set = 5 | The probability for an effectively contacted dog to immune-convert without overt disease. The published value is zero as this is not currently considered an important factor in rabies epidemiology. Low values were chosen to investigate the possible importance of such occurrences (if true). The default value was chosen as 0. |
| **initial incidence** | {0.003; 0.01; 0.03} Set = 3 | The epidemic is started by introducing for 2 consecutive months infected dogs equal to the annualised incidence. The maximum annual incidence (0.03) is twice reduced by a factor of 3 to produce the parameter space. The default value was chosen as 0.01. |
| **initial immunity** | {$0 \leq x \leq 1$} Set = 11 | The herd immunity at the start of each simulation (as a result of vaccination). Vaccinations are assumed to be 100% effective. All possible levels of herd immunity were investigated. The default value was chosen as 0.3 (a value typical of low intervention populations). |

[a] These model parameters were also used as the independent variables in the regression model.

This dataset was then analysed using a logistic regression model to identify the conditions that are most likely to result in rabies elimination. The prediction error of the regression model was minimized while controlling for complexity using stepwise regression, based on the Akaike Information Criterion (AIC). Since the aim of the study was to evaluate the impact on the required coverage, all factors were examined for their interaction with vaccination. The transmission model and the regression model were both computed using R [18]. Finally, two examples were calculated comparing the different contact rates (infectivity) in N'Djaména, Chad and comparing investment in sterilizations versus vaccinations in Hluvukani, South Africa.

## The disease transmission model

We made use of a peer-reviewed, publicly available, individual-based, stochastic model (http://github.com/anderaa/BioEcon) [16]. We retained the values for all parameters of the published form of the model except for our seven chosen study variables that were manipulated from a chosen minimum to a maximum (Table 1). The adapted R script used in this study is reported in S2 Script. The model used daily time steps and tracked the age, sex, reproductive status, immunity and disease status of individual dogs. Each of the variables took one of five different values within the chosen limits, apart from immunity and initial incidence. Immunity had eleven levels because a high level of precision was desired for interpretation over all possible values from 0 to 100% herd immunity. Initial incidence took on only three values because we wanted to compare the difference during maximum size outbreaks found under hyperendemic conditions versus very low incidences found when reintroductions of infective dogs occur in previously rabies-free areas.

In the model, the rate of effective bites from an infective dog varied stochastically following an evidence-based, negative binomial distribution that results in frequency-dependent transmission of disease [4]. Density-dependent transmission is often used instead, but there is empirical evidence in favour of frequency-dependent transmission. The issue remains controversial [8,14,19–21]. If the true transmission rate is indeed a function of density, it would be an important factor determining epidemic outcome by increasing the number of bites from each rabid dog [15]. However, other local ecology factors also directly impact the bite rate and includes fences, dog restraining devices and home ranges. It is therefore simpler to model the bite rate which varies as a function of the total dog ecology that includes dog density. Dog density is further confounded by the fact that interventions that reduce density also impact dog turnover rate, which in turn affects the decline of herd immunity following a vaccination pulse. We included a multiplier of the bite rate as a variable and called it infectivity.

Fecundity is defined as the reproductive potential of an animal and can be controlled through sterilizations. Reduced fecundity equates to reduced population turnover rates, meaning that herd immunity through vaccination is sustained for longer [22]. Although frequently applied as an adjunct to vaccination campaigns, its efficacy has been questioned [1,20,22]. Fecundity was also varied by using a multiplier affecting both the litter size and the number of litters.

We included a variable for the probability of a dog to immune-convert rather than develop disease following an effective exposure. Exposure acquired immunity is a critical determinant of epidemic outcomes in high-recovery diseases such as smallpox [23]. Some rabies challenge experiments have reported high rates of survival of non-vaccinated dogs [24,25]. However, field observations by Tepsumethanon et al. of large sets of biting but non-rabid dogs as well as true rabid dogs have failed to confirm this occurrence [26]. We therefore assumed that, if present under field conditions, the rate would be lower than the sensitivity of the field observations

by Tepsumethanon et al. (one in a thousand). We wanted to investigate if such events, even at low rates, impact the effectiveness of other control measures. We selected a low maximum probability to survive rabies challenge of 0.001. All survivors were non-infective and subsequently became immune.

The persistence of rabies has been reported to be unstable and reliant on frequent reintroductions from larger meta-populations [27–29]. Recently, a different stochastic model has found population size to be a highly significant predictor of elimination events [30]. Therefore, we included the initial population size and allowed it to vary over a large range (200 to 16,200). While rabies is often endemic in rural and remote regions where village sizes are smaller, very small population sizes are likely to result in mortality related population extinctions [19].

Substantial migration rates have been reported for dogs in Africa, where the typical in-migrant is not vaccinated [12]. We therefore classified all in-migrants as immune-naïve. We set the maximum in-migration at 10% per annum to preserve most of the original population characteristics, although higher rates have sometimes been reported in the field [12].

Migration is often associated with (re-)introductions of rabies [29]. This would represent the start of an epidemic with minimal incidence initially (before the epidemic gathers momentum). In mature epidemics with no interventions, maximal incidence can be expected. We varied initial incidence such to represent both these scenarios (as well as an intermediate value). At the lowest incidence value a population of 5,400 had two rabid dogs introduced for two consecutive months (all smaller populations had only one introduction for two months). At the highest incidence, a population of 5,400 had 14 rabid dogs introduced for two consecutive months.

Finally, as a seventh factor to examine, we varied initial herd immunity of the population between 0 and 1.

Each unique combination of parameter values (covariate pattern) was simulated 5 times each, producing a total of 515,625 iterations. From the authors' previous work using Anderson et al.'s model this was considered a large enough dataset to achieve adequate convergence. The first year of the simulation presents the highest probability for elimination due to the natural decline in immunity with time. At the end of one year, if no infectious or exposed individuals were present in the population at the simulation end-point, an elimination success (i.e. 1) was recorded. If the population of dogs itself had become extinct then that simulation result was discarded, thereby excluding results that were not necessarily related to virus extinction. Excluded data are reported in S1 Table. Such an event occurred at a rate of 0.009% over all parameter values, reaching 2.15% when the population size was set at the minimum value, 3.1% when the infectivity was set at a maximum and 9.4% when there was no herd immunity. Small population sizes, high infectivity rates and low herd immunities pre-disposed the model to extinction events and are over-presented in the data excluded from the analysis. No extinction events occurred when initial immunity was set at 0.1 or higher.

## The regression models

We calculated for each unique parameter combination (covariate pattern), the proportion of the iterations that resulted in successful elimination. The R script to do this is made available in S3 Script. We then modelled this proportion using a fractional logistic regression model and included the seven targeted parameters as covariates in a linear model (Model 1). Simple scatterplots of several of the independent variables revealed curvilinear relationships with elimination (S1 Fig). To account for potential non-linear relationships, a quadratic term was added for each independent variable (Model 2). Some of the extrema of the estimated quadratic equations fell just inside the parameter space, so we also investigated the inclusion of cubic terms

**Table 2. Derivation of the optimal regression model.**

| Model | Number of terms | Null deviance (df) | Residual deviance | AIC |
|---|---|---|---|---|
| Model 1 (linear model) | 8 | 473,847 (514,741) | 209,186 (514,734) | 209,202 |
| Model 2 (quadratic) | 15 | 473,847 (514,741) | 183,147 (514,727) | 183,177 |
| Model 3 (quadratic and cubic) | 21 | 473,847 (514,741) | 180,083 (514,721) | 180,125 |
| Model 4 (interactions) | 29 | 473,847 (514,741) | 191,782 (514,713) | 191,840 |
| Maximum model | 192 | 473,847 (514,741) | 167,678 (514,550) | 168,062 |
| Optimal model | 70 | 473,847 (514,741) | 167,368 (514,672) | 167,508 |

(Model 3). A fourth alternative model was investigated adding first order interaction terms for each of the independent variables with polynomial terms (Model 4). The four models were compared by their residual deviance and AIC. (Table 2). It was clear from the reductions in the residuals and drop in the AIC that quadratic terms, cubic terms and interactions all contributed to improving the fit of the model whilst controlling for complexity. From this we sought the optimal model by backward and forward stepwise regression of the maximum model using the AIC as the inclusion criterion. We chose AIC as the evaluation metric because it rewards superior fit and penalizes model complexity, which minimizes the risk of overfitting [31]. The model selection process was implemented with the stepAIC() function from the package MASS [32]. A script executable in R to define the regression models is available in S4 Script. The R summaries of all intermediate models (including each step of the stepAIC function) are available in S1 Text.

Each independent variable was plotted against the probability for stochastic elimination while holding all other variables (except for immunity) to their default values (refer to Table 1). Immunity was varied using 10% increments producing multiple plots. The predicted values from the regression model (predicted) were shown as well as the actual fraction of elimination events from the transmission model (observed). (Figs 1–6).

To explore the certainty of the value obtained for the elimination probability when the independent variables were adjusted, a second series of plots were created that show the frequency distribution of the elimination probability by means of boxplots (Figs 7–12). The wider the boxplots the smaller the confidence in the predicted probability for elimination. This would be as a result of the effect of the other independent variables.

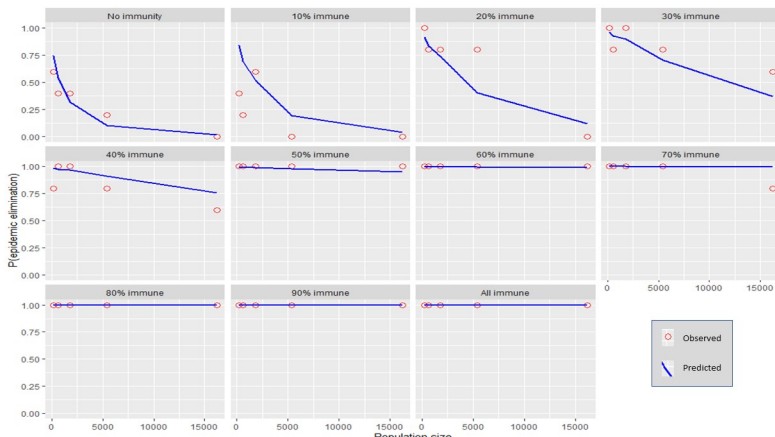

**Fig 1. The response of P(elim.) to population size at different herd immunities.** The proportion of elimination events from the transmission model (red circles) and the prediction from the regression model (blue line).

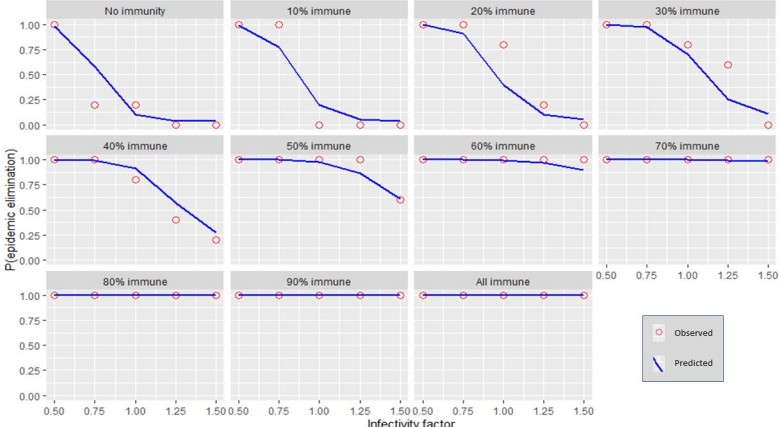

**Fig 2. The response of P(elim.) to the infectivity factor at different herd immunities.** The proportion of elimination events from the transmission model (red circles) and the prediction from the regression model (blue line).

A heatmap was made to explore the interactions between the variables identified as the strongest contributors to elimination (Fig 13). The intensity of the colour represented the probability of elimination. The initial incidence was varied on the y-axis, the initial population size on the x-axis and the infectivity was adjusted across the panels of the plot. Other variables were held at their default values.

A second heatmap was made to evaluate how the effect of fecundity may change across values of infectivity and population size (Fig 14). This was done to determine if specific ecologies may optimize the effect of fecundity.

The R script to produce all the Figs 1–14 is made available in S5 Script.

## Comparing different contact rates (infectivity) present in N'Djaména, Chad

A recent dog ecology study from N'Djaména, Chad, constructed a contact network through geo-located contact sensors. It was found that two adjacent, but culturally different suburbs

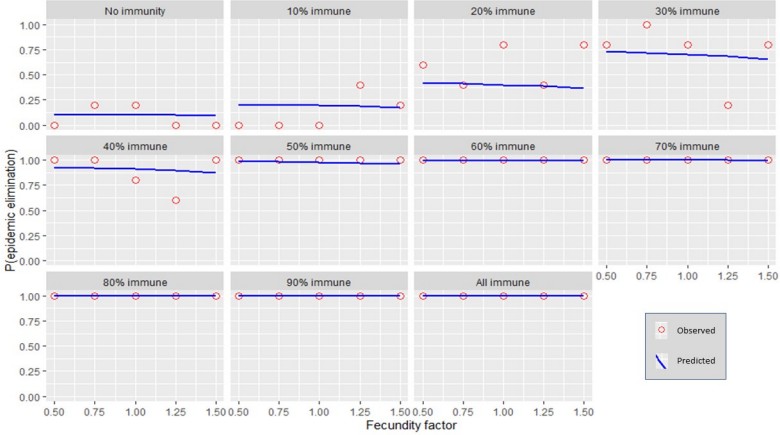

**Fig 3. The response of P(elim.) to the fecundity factor at different herd immunities.** The proportion of elimination events from the transmission model (red circles) and the prediction from the regression model (blue line).

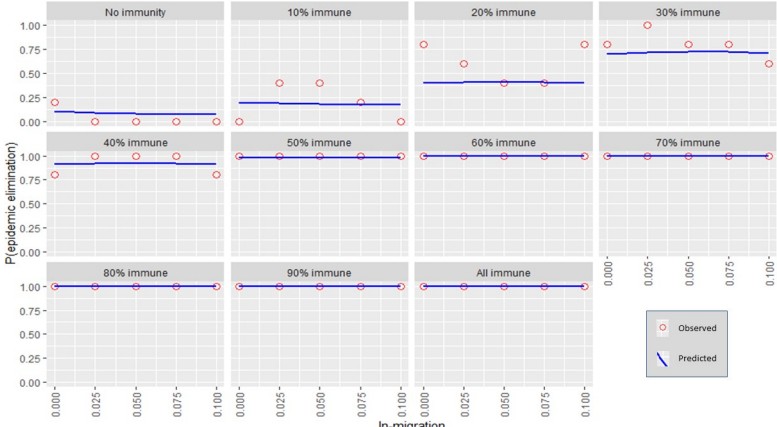

**Fig 4. The response of P(elim.) to the in-migration rate at different herd immunities.** The proportion of elimination events from the transmission model (red circles) and the prediction from the regression model (blue line).

within one city had different connectivity's equal to 15 and 9 respectively [32]. This is equal to a 40% difference in infectivity. We applied the same relative difference in infectivity to the model, reducing infectivity from 1 to 0.6 or increasing it from 1 to 1.4. We examined how these differences in infectivity effect the probability for elimination. We kept the other variables at their default values including immunity at 0.3 and population size at 5400 (which represented a typical large district of N'Djamena [33].

## Comparing the effect of investing in sterilizations or vaccinations

A scenario was considered based on a dog population of 16,200 (holding all other variables at their default value including immunity at 30% and fecundity at 1). The effect of sterilizing female dogs on the probability for elimination, P(elim.), was investigated by bringing the fecundity factor down from 1 to 0.75. This was compared to a second strategy where the same budget required for sterilization was instead applied to vaccinate dogs. In both scenarios the capture costs from the BioEcon model were accepted (this represents the logistical costs of getting access to a dog to perform an intervention). The capture costs increase in a stepwise linear fashion with four brackets representing increasing cost to capture dogs (0–25%; 26–50%;

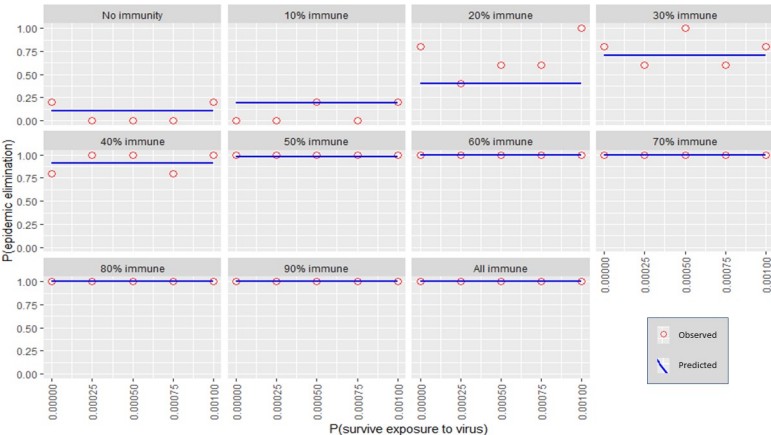

**Fig 5. The response of P(elim.) to exposure survival probabilities at different herd immunities.** The proportion of elimination events from the transmission model (red circles) and the prediction from the regression model (blue line).

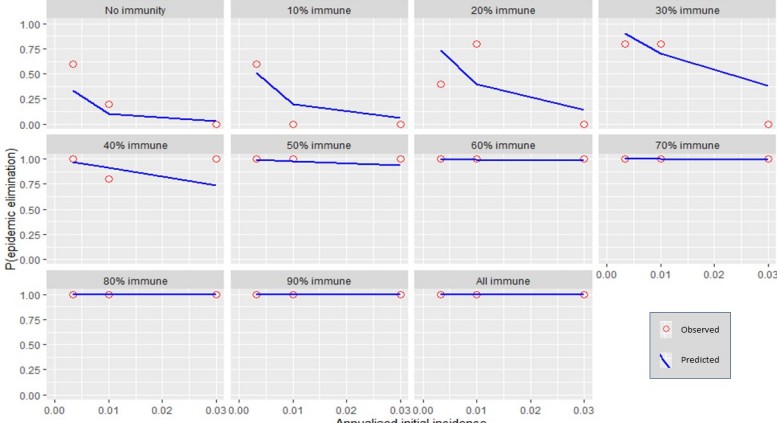

**Fig 6. The response of P(elim.) to initial incidence at different herd immunities.** The proportion of elimination events from the transmission model (red circles) and the prediction from the regression model (blue line).

51–75%;76–100%) [16]. This is a result of the increasing effort required to access progressively higher proportions of dogs in a population. The total capture costs were calculated based on how many dogs from each bracket required the intervention. The sterilization strategy focussed on bitches only, due to the assumption that a reduction of fertile males only effects population fecundity at very high proportions of male sterility. The vaccination strategy included puppies because it follows WHO recommendations for mass campaigns [1]. The procedural costs of vaccinations and sterilization were adopted from Anderson et al.'s BioEcon model [16]. The costs were adapted for the latest ZAR to USD exchange rate (R1.00 ZAR = $0.066 USD on 7/12/2020).

## Results

### Regression analyses

The final regression model (optimal model) contained 70 terms including cubic and quadratic interactions. The focus of the analyses of the regression model was therefore graphical

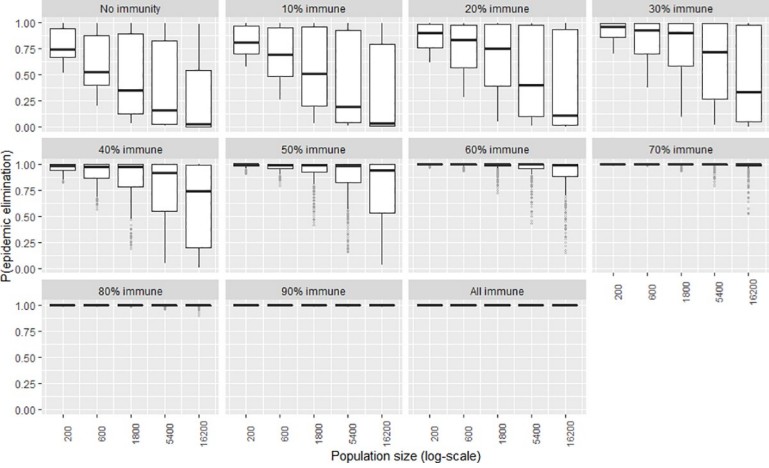

**Fig 7. The certainty of the effect of population size on P(elim.) at different herd immunities.** The range of possible values when other parameters not shown in this plot are varied over their ranges. The boxplot shows the median, the interquartile range, minimums and maximums (1.5 times the 1st and 3rd quarter values) and outliers (grey circles).

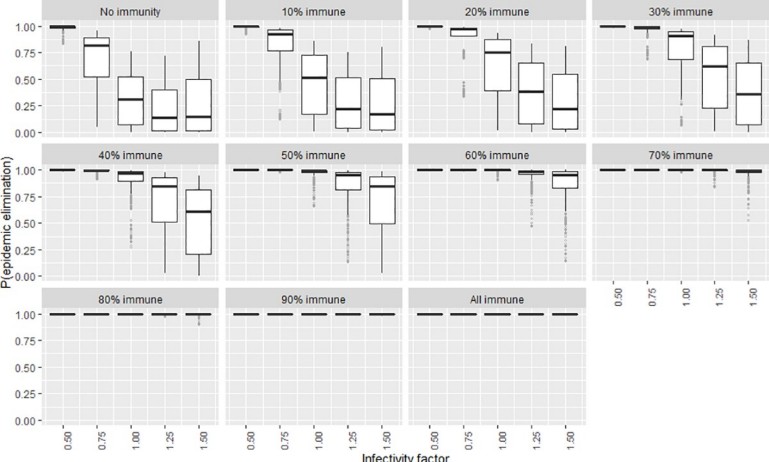

**Fig 8. The certainty of the effect of the infectivity factor on P(elim.) at different herd immunities.** The range of possible values when other parameters not shown in this plot are varied over their ranges. The boxplot shows the median, the interquartile range, minimums and maximums (1.5 times the 1st and 3 rd quarter values) and outliers (grey circles).

interpretation with selected outcomes highlighted under certain parameter conditions. The R-output containing the terms included in the final model are presented in S2 Text, with their coefficients and p-values. However, analysing coefficient p-values is of limited value when regression models are based on datasets that are outcomes of models themselves (in this case the transmission model) because the number of iterations chosen by the researcher directly affects the p-values.

## Initial population size

Generally, the probability for elimination, P(elim.) decreased exponentially as the population size increased (see Fig 1). The effect of population size was maximised at 10% immunity at

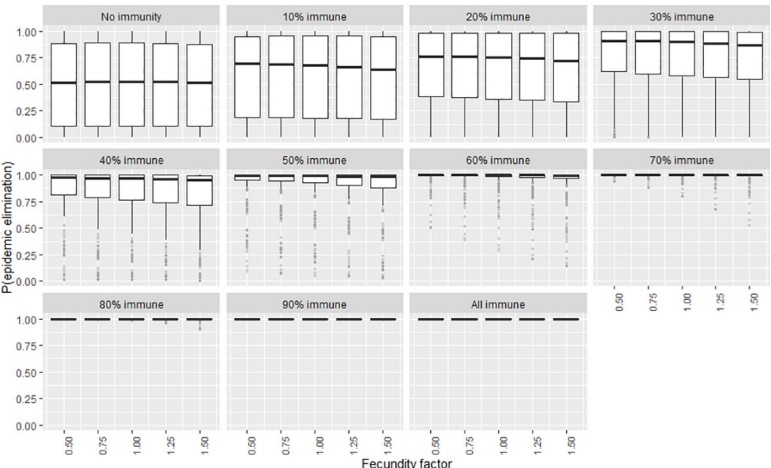

**Fig 9. The certainty of the effect of the fecundity factor on P(elim.) at different herd immunities.** The range of possible values when other parameters not shown in this plot are varied over their ranges. The boxplot shows the median, the interquartile range, minimums and maximums (1.5 times the 1st and 3 rd quarter values) and outliers (grey circles).

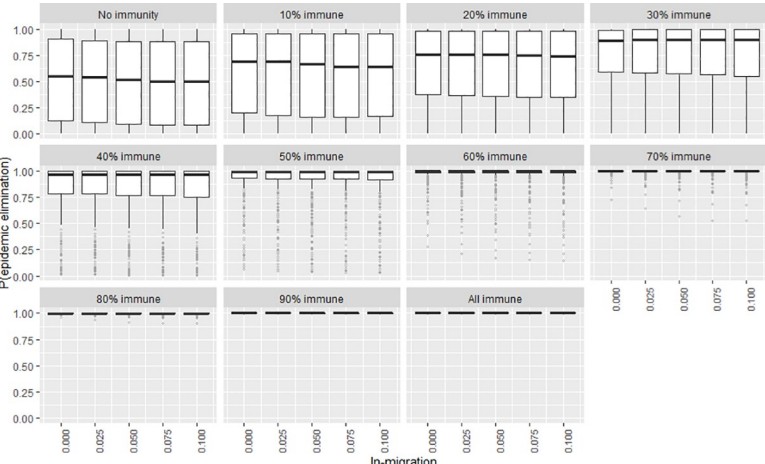

**Fig 10. The certainty of the effect of in-migration rates on P(elim.) at different herd immunities.** The range of possible values when other parameters not shown in this plot are varied over their ranges. The boxplot shows the median, the interquartile range, minimums and maximums (1.5 times the 1st and 3rd quarter values) and outliers (grey circles).

which increasing population from 200 to 16,200 resulted in a decrease of P(elim.) from 0.84 to 0.04. Population sizes of 600 or less could reliably affect elimination (>90%) when other independent variables were held at their default values. Elimination at small population sizes was less dependent on the values of the other independent variables (Fig 7. Notice the increasing sizes of the interquartile ranges as population size increases).

## Infectivity

At low immunity levels (0–40%) the effect of infectivity was maximised, following an inverse sigmoid shape as the infectivity factor increased (Fig 2). At higher vaccination levels (immunity >40%), P(elim.) approached 1 regardless of the infectivity. Infectivity factors of 0.75 or

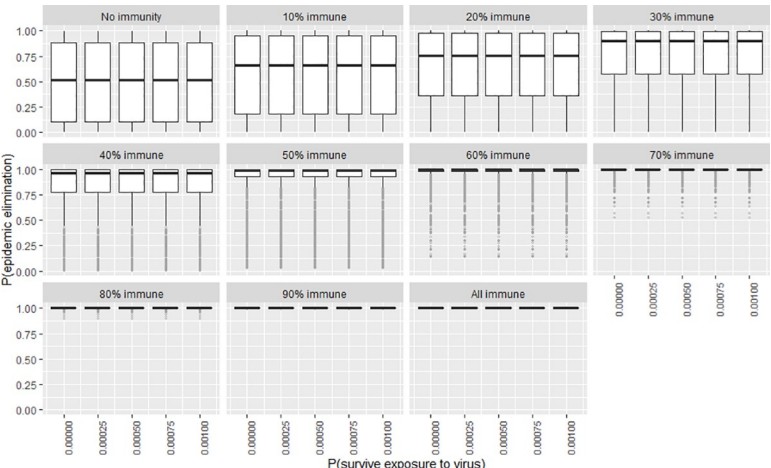

**Fig 11. The certainty of the effect of exposure survival on P(elim.) at different herd immunities.** The range of possible values when other parameters not shown in this plot are varied over their ranges. The boxplot shows the median, the interquartile range, minimums and maximums (1.5 times the 1st and 3rd quarter values) and outliers (grey circles).

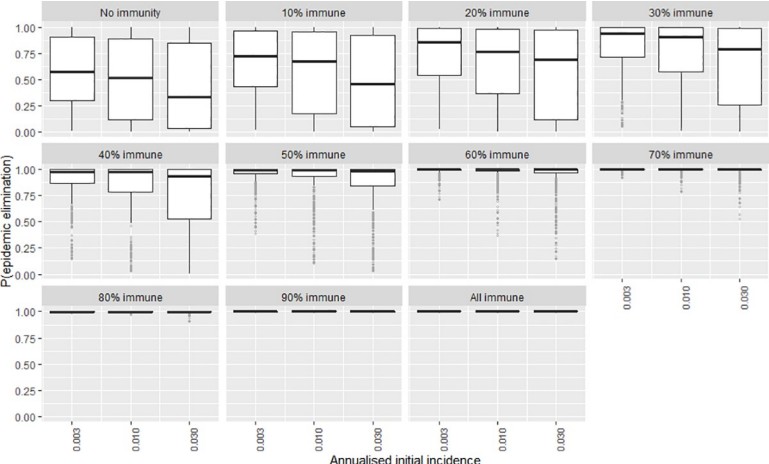

**Fig 12. The certainty of the effect of initial incidence on P(elim.) at different herd immunities.** The range of possible values when other parameters not shown in this plot are varied over their ranges. The boxplot shows the median, the interquartile range, minimums and maximums (1.5 times the 1$^{st}$ and 3d quarter values) and outliers (grey circles).

less could reliably produce elimination (>90%) when other independent variables were held at their default values. When reducing infectivity further to 0.5 elimination became almost certain regardless of the value of all the other independent variables, including even the herd immunity. This meant that after initial immunity, infectivity was the least dependent on other independent variables to produce high probabilities for elimination (this is apparent from the narrow bands of P(elim.) outcomes at low infectivity) (Fig 8). It appeared that where dog contact rates could be lowered by more than 25% (infectivity < 0.75), the population size became less important (Fig 13).

## Fecundity

A weak linear decrease in P(elim.) could be observed as fecundity increased (Fig 3). The small effect of fecundity was overshadowed by the stronger effect of other independent variables

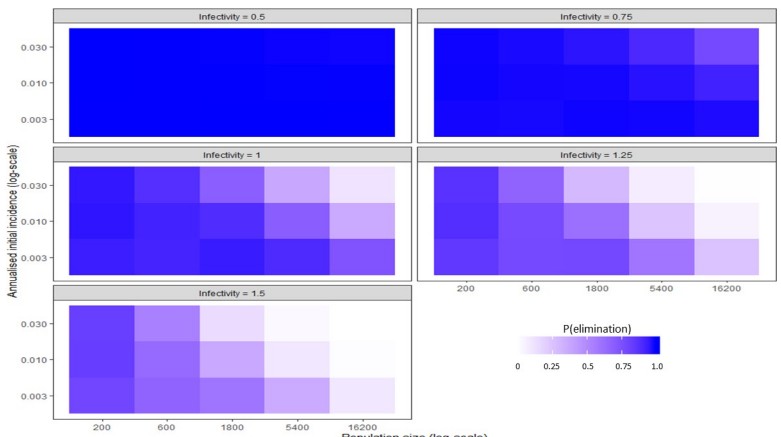

**Fig 13. A heatmap showing the effect of population size, infectivity and initial incidence on P(elim.) at 30% herd immunity.**

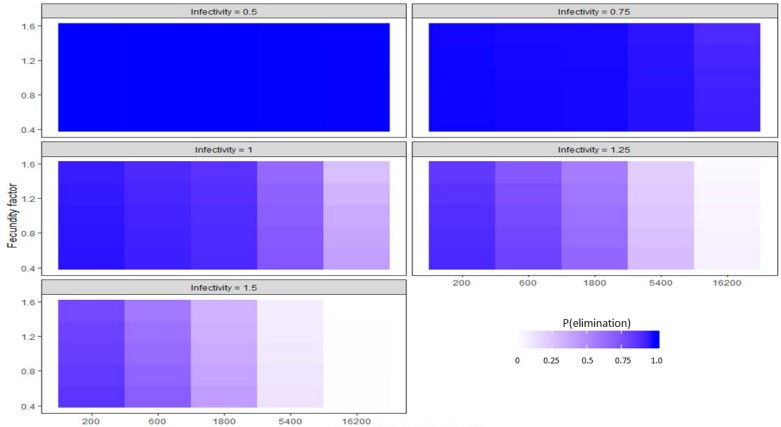

**Fig 14. A heatmap showing the effect of population size, infectivity and fecundity on P(elim.) at 30% herd immunity.**

(Figs 9 and 14) but was nevertheless maximised at 30% immunity where an increase in the fecundity factor from 0.5 to 1.5 resulted in a decrease in P(elim.) of 7.8%. We could not detect clear evidence that the effect size of fecundity changed substantially at different levels of population size or infectivity (Fig 14).

## In-migration and exposure survival

The effect of in-migration and exposure survival was mostly insubstantial at the observed ranges (although significant enough to be included in the model for accuracy purposes) (Figs 4–5). Changes in these variables also did not substantially change the spread of P(elim.) as did other significant variables (Figs 10–11).

## Initial incidence

At low immunity levels (0–40%) P(elim.) decreased substantially as annual incidence increased (Fig 6). At higher vaccination levels (immunity >40%), P(elim.) approached 1 regardless of the initial incidence. At 20% immunity the effect of initial incidence was maximised. An increase in initial incidence from 0.003 to 0.03 drastically reduced P(elim.) from 0.74 to 0.14. Initial incidence of 0.003 or less (typically representing single-dog introductions) resulted in a P(elim.) >90% when other independent variables were held at their default values. This effect was less reliable when other variables could vary within their parameter space (Fig 12). Ecologies that combined a low initial incidence (small, new outbreaks) with low infectivity (low contact rates between dogs) and small sub-populations maximised the probability for elimination (Fig 13).

## Initial immunity

Immunity levels of 40% or more had a P(elim.) >90% when the other independent variables were held at their default values (Figs 1–6). Immunity levels of 70% and above almost invariably led to epidemic elimination regardless of the values allocated to other independent variables (Figs 7–12). Low levels of immunity could result in high probabilities of elimination under ecological conditions of small populations, low infectivity and low initial incidence (Figs 1–2 and 6).

**Table 3. Marginal costs to raise P(elim.) through vaccination or sterilization.**

| | Sterilization | Vaccination |
|---|---|---|
| Baseline population before intervention is **16,200 dogs** that is **30% immune** irrespective of demographics. | | |
| 61% are adult dogs of which 38% are female. **Fecundity factor is 1**. | | |
| Number of dogs eligible for procedure: | 16,200 x 0.61 x 0.38 = 3,755 | 16,200 |
| Number of dogs already vaccinated/ sterilized: | 0 | 30%; 16,200 x 0.3 = 4,860 |
| Number of dogs to receive procedure: | 939 (0–25%) | 11,340 (30–100%) |
| Capture costs are $0.58; $0.99; $1.13; $2.22 per dog in the population brackets 0–25%; 26–50%; 51–75%; 76–100% | | |
| Costs to capture dogs for interventions: | $544.62 | $15,913.98 |
| | = $0.58 x 939 | = ($0.58 x 0) + ($0.99 x 3235) + ($1.13 x 4050) + ($2.22 + 4050) |
| Procedural costs are $19.80 per bitch sterilization and $0.16 per dog vaccination. | | |
| Total procedural cost for all dogs: | $18,592.20 | $1,814.40 |
| | = $19.80 x 939 | = $0.16 x 11340 |
| Total intervention cost: | $19,136.82 | $18,589.10 |
| Demographic effect | Fecundity moves from 1 to 0.75. | Herd immunity moves from 0.3 to 1. |
| P(elim.) before intervention: | 0.37 | 0.37 |
| P(elim.) after intervention: | 0.40 | 1.00 |
| USD[a] spent per percentage point gained in P(elim.): | $6,378.94 | $295.07 |

## Comparing different contact rates (infectivity) present in N'Djaména, Chad

With the baseline infectivity at 1, the modelled population had a P(elim.) of 0.70. Decreasing infectivity by 40% resulted in a P(elim.) rounded to 1, whereas increasing infectivity by 40% resulted in a reduction of P(elim.) to 0.14.

## Comparing the effect of investing in sterilizations or vaccinations

The total cost of reducing fecundity from 1 to 0.75 in a population of 16,200 dogs through bitch sterilizations was $19,136.82 (Table 3). Most of the costs were due to high procedure costs ($18,592.20) with low capture costs ($544.62). This strategy increased the probability for elimination 37% to 40% (a cost of $6,378.94 per percentage point increase). The total cost of raising the herd immunity from 30% to 100% was less than for the sterilization strategy at $18,589.10. In this strategy the procedural costs were low ($1,814.40) and the capture costs high ($18,589.10). The vaccination strategy increased the probability for elimination from 37% to 100% (a cost of $295.07 per percentage point increase).

## Discussion

### Initial population size

At any point in time, the probability for the elimination of rabies in a dog population is the product of the individual probabilities of zero effective secondary contacts of all existing infected dogs. Such joint probabilities (of any stochastic event) will always be more likely the fewer probability events there are (in our case, the fewer infected dogs there are). Small dog populations will have fewer infected dogs than large populations and are therefore more likely to undergo stochastic extinctions. Secondly, dog rabies is a rare disease with published

maximum annual incidences of 0.25% [4] or 3% [34]. This means that the number of infected dogs on whom the propagation of the epidemic depends is often limited to a few individuals. Thirdly, the frequency distribution of secondary contacts from rabid dogs, approaches a negative binomial distribution with the occurrence of zero secondary contacts being common [4].

This paper provides further support from a modelling perspective that smaller dog populations have higher probabilities for elimination [30]. Empirically there is evidence that rabies virus lineages go extinct in sub-populations [33], independent of control measures [28]. It has been previously stated that rabies survival is dependent on large meta-populations [27–29,35], but the magnitude of the impact of small, sub-populations is not yet fully realised or applied. In mathematical biology, however the importance of population size to ensure survival is well known and described [36,37].

The critical population size required to sustain rabies quickly enlarges when infectivity is only slightly reduced (Fig 13). Campaign managers could implement this knowledge through two interventions. They could attempt to fragment large metapopulations into progressively smaller sub-populations by vaccinating corridors to prevent virus passage between the sub-populations. Such corridors can be selected to augment existing barriers that inhibit dog movement, thereby further reducing the required resources. The topography of some endemic zones lend themselves to obvious fragmentation with rivers and high mountains, but even busy roads can effectively restrict dog movement [33]. The second approach would be to identify sub-populations that are unable to sustain rabies epidemics by themselves. The campaign manager can then withdraw resources from such sub-populations and focus them elsewhere. The design of the strategy should aim to maximize the width of immune-cordons and minimize the size of the divided sub-populations.

Human mediated dog movements are often reported to have resulted in the spread of rabies over long distances [29]. This can be an especially important means of introduction deep into rabies free zones. However, this fact should not detract attention from the main route of transmission which is by local dogs on foot. By far the majority of secondary contacts from rabid dogs occur within 1km from the source [4].

To put it in terms of network analysis: campaign managers should seek to target those nodes that have high betweenness centrality to create disassociated sub-populations. This means they should target sub-populations that more often act as bridges to connect two other sub-populations [38]. A proviso for such an approach would be that those sub-population sizes are below the critical threshold to sustain rabies. Such critical thresholds will not be the same everywhere, especially where contact rates (i.e. infectivity) differ.

## Infectivity

Infectivity is a pathogen's ability to be transmitted. Rabies is spread through direct contact making it directly proportional to infectivity. Whereas some rabid dogs have higher contact rates than healthy dogs, the rabid dog's contact rate remains a function of the local dog ecology. In simple terms, a rabid dog from a setting where dogs roam freely will find it easier to effectively contact other dogs than one from a society where dogs are locked behind solid walls. The N'Djamena example illustrates that baseline infectivity rates can differ substantially even in the same city (40%) and have substantial impact on the local probabilities for elimination. Because the districts of N'Djamena are geographically adjacent with long, common borders, the districts should not be viewed as distinct sub-populations. Therefore, we do not specifically advise to neglect the districts in N'Djamena with lower contact rates during mass vaccination campaigns. We do however believe that there are many real-world settings where such differences in contact rates exist and where the sub-populations are somewhat isolated.

In such circumstances our results suggest that campaign managers should prioritise resources to the sub-populations with higher contact rates. Alternatively, sub-populations with very low infectivity factors that also have high betweenness centrality may present opportunities to separate meta-populations. Using minimum resources to vaccinate these strategic sub-populations will in turn increase the probability for elimination in the sub-populations connected to them by reducing their effective population sizes.

Reducing infectivity through interventions may be difficult in some settings. Tie-up orders (although resource intensive) will be very effective. Educating the public to restrict free roaming behaviours and cleaning up or fencing rubbish dumps (where contact rates are typically very high) will also help. Herein lies a worthwhile opportunity to involve local authorities from sectors other than veterinary or human health.

## Fecundity

Higher fecundity increases the population turnover, thereby depleting herd immunity faster. For this reason, fertility control has been advocated as an adjunct to vaccination in rabies campaigns [39]. Most commonly, fecundity is controlled through surgical sterilizations. Importantly though, such sterilizations should not undermine resources allocated to vaccination [1]. Our results reaffirm this advice.

The apparent weak effect of fecundity should not be completely discarded by campaign managers. Firstly, many donors insist on sterilization components to control programs, and their funds would otherwise be lost if not applied [40]. Secondly, some higher income sub-populations already have higher sterilized proportions of dogs due to owner interventions unrelated to rabies control. These same sub-populations often have lower bite rates (infectivity) due to better fences. Such sub-populations will require lower intensity vaccination campaigns to achieve elimination and should therefore be a lower priority for resource constrained campaign managers.

## In-migration

In this study, in-migration had a minimal impact on the elimination probability. It is however important to remember, that the in-migrant dogs were non-infective and immune-naïve. The phenomenon where in-migrants introduce rabies into rabies free areas has been examined using the initial incidence variable (low incidence represents a single introduction). Campaign managers should attempt to limit in-migration for this reason. Sub-populations where in-migration rates are higher than 10% per annum should not be viewed as sub-populations as defined in this paper.

## Exposure survival

Surviving exposure could impact rabies epidemics by reducing the number of effective contacts that lead to disease and by increasing herd immunity as the exposed dogs immune-convert. It appears that at a rate below 1 in a 1,000 this impact is negligible. If future research confirms survival after effective exposure in dogs is a more common event, then the impact thereof will have to be reviewed.

## Initial incidence

The initial incidence multiplied with a transmissibility constant (in our case effective bites) gives the force of infection [41]. The greater the force of the infection the harder it is to stop it. In the model, initial incidence represents the number of cases at the start of the iteration. In

the field it represents the maturity of the epidemic. Where rabies epidemics have been allowed to continue unabated the elimination of rabies will present a greater challenge requiring higher herd immunities. Whereas keeping rabies-free area free of rabies is vitally important, campaign managers should be aware that the required herd immunity to do that is lower than the required herd immunity to eliminate an active epidemic [42]. Another important implication is that maintaining herd immunity at a certain level for a fewbite years will not only increase the probability for elimination due to a prolonged window of opportunity, it will also reduce the force of the infection making the elimination more likely in successive years (due to reducing incidence). Where campaign managers are using reported incidence to inform their strategy, they need to carefully consider the relative sensitivity of the surveillance system.

## Initial immunity

It is clear from the results that raising herd immunity is the most effective way to control dog rabies. More so since vaccination is the most cost-effective intervention (Table 3). Vaccination campaigns offer more potential benefits that were not considered in this analysis. For one, a vaccination campaign in itself can be an awareness campaign (whether preceded by awareness campaigns or not).

After vaccination campaigns are applied, the herd immunity immediately starts to decline due to the turnover of the population and the natural waning of immunity in the individual. Capaign managers should therefore be careful about the assumptions they make on the current herd immunity. We recommend that campaign managers should annually re-evaluate their position with respect to elimination probabilities by re-defining all of the population parameters. This should include a downward adjustment of the expected immunity from previous campaigns to a current population immunity. We provide a possible formula with $I_t$ representing the estimated herd immunity at any time $t$. $I_t = I_{t=0} \cdot \left(1 - \frac{1}{v} - \frac{1}{p}\right)^t$ where $v$ is the expected duration of vaccine immunity in years, $p$ is the proportion of the population replaced annually (due to births, deaths and dispersal) and $t$ is the time since the last campaign in years.

## Limitations

Our recommendations are based on identifying the specific ecologies of different sub-populations and adapting the control measures accordingly. This assumes that the sub-populations are isolated at least to some extent. The less isolated the sub-populations are the less likely our findings are to hold true. It is our opinion that most rabies-endemic areas do have natural constrictions in the connectivity of the underlying sub-populations. It is up to the local campaign manager to identify where these constrictions are and to what extent they result in relative isolation. Conversely, it may often be the case that a series of villages are so well connected as to present only one sub-population.

Each simulation was only run for one year. Yet, field experience shows that campaigns to eliminate rabies generally need to be sustained for at least a few years. We have opted not to include an analysis of model results beyond 1 year because we have found that the repeatability of predictions from our model declines rapidly the further it predicts into the future. Early random events in stochastic models lead to high uncertainty in distant predications, sometimes called a *butterfly effect*. It is therefore better for campaign managers to re-establish the expected probabilities for elimination and their subsequent strategies at least annualy. This will optimise the precision achievable from a stochastic model.

The parameters for the transmission model were adopted from Anderson et al.'s bioeconomic model. The model draws heavily on data collected in the Ehlanzeni region of

Mpumalanga in South Africa and on Hampson et al.'s transmission dynamics parameters [4,16]. Whereas the 7 study variables were varied across large ranges, other parameters of the BioEcon model were held constant. This may affect the external validity of the observations made in this study, but it is our view that it is unlikely to change the direction of observed effects. This is especially true where the observations are linked to cost calculations (Table 3).

## Conclusions

Epidemics are by nature stochastic. Control measures cannot ensure epidemic elimination but rather push the stochastic processes in a direction that increases the likelihood of elimination. Strategic allocation of resources that considers the host ecology and the local disease epidemiology can be optimised if better estimates are available to predict local elimination probabilities. Dog ecologies differ substantially between rabies endemic areas, and one strategy cannot fit all situations [14]. The only constant, it seems, would be that vaccination remains the most effective way to control rabies. Keeping in mind that uniform coverage of areas with vaccination coverage of at least 70% remains the surest route to success, campaign managers that are unable to do so should strategically apply the resources they have. We have identified some factors that can guide such decisions. The most important factors advancing local eliminations are smaller populations, lower contact rates (infectivity) and lower incidence. This paper highlights the potential advantages for local campaign managers to understand the local dog ecology, especially as it pertains to dog contact rates within and between sub-populations.

## Supporting information

**S1 Table. Excluded iterations.** The parameter values for each excluded iteration due to population extinction events.
(CSV)

**S1 Fig. Scatterplots.** The model parameters plotted against the probability for elimination.
(PDF)

**S1 Script. R script for in-text calculations.** Executable in R.
(R)

**S2 Script. R script for transmission model.** Executable in R as a standalone script.
(R)

**S3 Script. R script to rearrange data for regression models.** Executable in R.
(R)

**S4 Script. R script for regression models.** Executable in R.
(R)

**S5 Script. R script for figures.** Reproduce the figures. Executable in R.
(R)

**S1 Text. Alternative regression model results.** The regression coefficients, standard errors and p-values for the optimal model.
(LOG)

**S2 Text. Optimal model results.** The regression coefficients, standard errors and p-values for the optimal model.
(TXT)

## Author Contributions

**Conceptualization:** Johann L. Kotzé, Aaron Anderson.

**Data curation:** Johann L. Kotzé.

**Formal analysis:** Johann L. Kotzé.

**Funding acquisition:** Johann L. Kotzé.

**Investigation:** Johann L. Kotzé.

**Methodology:** Johann L. Kotzé, Aaron Anderson.

**Project administration:** Johann L. Kotzé.

**Resources:** Johann L. Kotzé, John Duncan Grewar, Aaron Anderson.

**Software:** Johann L. Kotzé, Aaron Anderson.

**Validation:** Johann L. Kotzé.

**Visualization:** Johann L. Kotzé, John Duncan Grewar.

**Writing – original draft:** Johann L. Kotzé.

**Writing – review & editing:** Johann L. Kotzé, John Duncan Grewar, Aaron Anderson.

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
