## [Decision Letter · Decision Letter 0]

8 Apr 2020

Dear Dr Kotze,

Thank you very much for submitting your manuscript "Modelling the factors affecting the probability for local rabies elimination by strategic control" for consideration at PLOS Neglected Tropical Diseases. As with all papers reviewed by the journal, your manuscript was reviewed by members of the editorial board and by several independent reviewers. In light of the reviews (below this email), we would like to invite the resubmission of a significantly-revised version that takes into account the reviewers' comments. 

I thank the authors for this interesting contribution to an important and timely topic. If the aim is to advance the science behind the management of canine rabies control, then I find a serious limitation of the current work to be that the authors do not consider a scenario of disease re-introduction to the simulated population(s) and at least one other reviewer also questioned the validity of that assumption and whether it applies to the majority of dog populations where rabies virus circulation is enzootic. I also can imagine scenarios where managers may have little control over the immigration of infected dogs which could be driven more by human socioeconomics and behavior (e.g. compliance) regardless of governmental regulations in place – akin to the ‘societies’ the authors allude to in Discussion, so I struggle with how some of the recommendations are practical or also consider the logistic and economic challenges of introducing spatial edge effects with barrier vaccination strategies targeting animal populations. I recommend that additional experiments be carried out or additional references for justification supplied which do consider an event probability of disease re-introduction to managed dog populations. It seems this phenomenon is commonly observed with dog and wildlife rabies management programs in many places, especially but not only where jurisdictional management objectives, actions and coordination vary. 

The authors occasionally make statements that appear to generalize about the entire field of animal rabies or its control, where perhaps they should be more circumspect to specify dog rabies/control or otherwise reference and interpret theoretical principles and relevant empirical works regarding wildlife epizootiology and control (e.g. bats and carnivores). There are sparse reference to or mention of wildlife literature on these topics. 

All of the regression model testing and AIC comparison results must be made available and accessible – this information is not described anywhere in the results nor could I find it in the supplemental files. How many regression models were tested, the number of parameters, AIC, model weight and other relevant metrics should be described. Please also include some reference to null (intercept only) regression model testing. I recommend inclusion of these results in the main text rather than as supplementary material. 

Lastly, there are several instances of text/figures in inappropriate places (see reviewer comments for specific examples). I ask the authors to consider some changes to improve the organization and flow of information in the paper.

We cannot make any decision about publication until we have seen the revised manuscript and your response to the reviewers' comments. Your revised manuscript is also likely to be sent to reviewers for further evaluation.

Sincerely,

Amy T. Gilbert

Deputy Editor

Sergio Recuenco

Deputy Editor

I thank the authors for this interesting contribution to an important and timely topic. If the aim is to advance the science behind the management of canine rabies control, then I find a serious limitation of the current work to be that the authors do not consider a scenario of disease re-introduction to the simulated population(s) and at least one other reviewer also questioned the validity of that assumption and whether it applies to the majority of dog populations where rabies virus circulation is enzootic. I also can imagine scenarios where managers may have little control over the immigration of infected dogs which could be driven more by human socioeconomics and behavior (e.g. compliance) regardless of governmental regulations in place – akin to the ‘societies’ the authors allude to in Discussion, so I struggle with how some of the recommendations are practical or also consider the challenges of introducing spatial edge effects with barrier vaccination strategies targeting animal populations. I recommend that additional experiments be carried out or additional references for justification supplied which do consider an event probability of disease re-introduction to managed dog populations. It seems this phenomenon is commonly observed with dog and wildlife rabies management programs in many places, especially but not only where jurisdictional management objectives, actions and coordination vary. 

The authors occasionally make statements in a way that seems to generalize about the entire field of animal rabies or its control, where perhaps they should be more circumspect to specify dog rabies/control or otherwise reference and interpret theoretical principles and relevant empirical works regarding wildlife epizootiology and control (e.g. bats and carnivores). 

All of the regression model testing and AIC comparison results must be made available and accessible – this information is not described anywhere in the results nor could I find it in the supplemental files. How many regression models were tested, the number of parameters, AIC, model weight and other relevant metrics should be described. Please also include some reference to null (intercept only) regression model testing. I recommend inclusion of these results in the main text rather than as supplementary material. 

Lastly, there are several instances of text/figures in inappropriate places (see reviewer comments for specific examples). I ask the authors to consider some changes to improve the organization and flow of information in the paper.

Reviewer's Responses to Questions

**Key Review Criteria Required for Acceptance?**

**Methods**

-Are the objectives of the study clearly articulated with a clear testable hypothesis stated?

-Is the study design appropriate to address the stated objectives?

-Is the population clearly described and appropriate for the hypothesis being tested?

-Is the sample size sufficient to ensure adequate power to address the hypothesis being tested?

-Were correct statistical analysis used to support conclusions?

-Are there concerns about ethical or regulatory requirements being met?

Reviewer #1: Line 112-125: I understand the struggle between frequency-dependent transmission and density-dependent transmission, esp. when there is indication that rabies transmission may be a mixture of both; at high densities density-dependent transmission may become more important than at low densities. Also, the parameter ‘fecundity’ is density-dependent (high densities – smaller litter sizes and/or reduced number of reproductive active vixens) although due to human interference this is less pronounced with domestic dogs than with wildlife canid species. I struggle with the fact that the authors only consider the absolute population size and not the density of the dog population as the latter can impact disease transmission.

Line 158- 171: what about interactions, there are not mentioned here at all but according to Table S2 they were (partially?) included in the model. So, please include a statement on the incorporation of interactions. However, it seems only first order interaction (A*B, B*D, etc.) and not higher order interactions (A*B*C or A*C*D*E, etc.) were investigated; please clarify why, especially considering incorporating cubic terms.

Line 174: The final model included 76 terms incl. cubic and quadratic interactions. To be honest, I feel uncomfortable with such models. To include so many terms, makes it almost uncontrollable. Instead of focussing on the AIC there is more simplistic way: Start with a linear model incl. first order interactions. Sometimes this is not possible for all interactions because of lack of degrees of freedom. Perform a residuals analysis and see if you need to adapt you model by including certain quadratic terms, if it is clear that for a certain factor the addition of a quadratic term improves the predictability of the model. You can further ‘polish’ your model by removing certain factors and interactions if they have no significant effect. The reference in line 177 to Table S1 is incorrect, the model terms are shown in Table S2. 

Line 173-244: Should the analysis with all the figures not be part of the Results. The model selected is based on decisions of the authors, so the figures provided are results obtained.

Reviewer #2: Further detail regarding certain results presented in the Discussion are required

Reviewer #3: Line 135-Although they remain controversial and are difficult to replicate, there are several studies suggesting 'naturally acquired' immunity in free roaming dogs (e.g Millan 2013, 10.3201/eid1904.121143). The inclusion of naturally acquired immunity is therefore a good idea, but it could even be higher if those few studies are to be believed.

The authors should explain why the model only runs for one year. It is beyond the expectations of most campaign managers to eliminate the disease in a year

Line 124- given that puppies are dependent on their mothers for a period, was the model able to account for puppies that would die as a result of the mother being infected?

**Results**

-Does the analysis presented match the analysis plan?

-Are the results clearly and completely presented?

-Are the figures (Tables, Images) of sufficient quality for clarity?

Reviewer #1: I think as stated previously that the figures should be part of the results.

Concerning data availability; the terms selected for the model can be found in Table S2 but what about the terms not included. What are the criteria for omitting certain (interaction) terms?

Reviewer #2: Further detail regarding certain results presented in Fig 11 and the Discussion are required

Reviewer #3: Results are clearly presented

**Conclusions**

-Are the conclusions supported by the data presented?

-Are the limitations of analysis clearly described?

-Do the authors discuss how these data can be helpful to advance our understanding of the topic under study?

-Is public health relevance addressed?

Reviewer #1: See, general comments. I think the conclusions could could be made using common sense and that this overcomplicated model gives the hard data presented (population size and subsequent probability of elimination with a fixed number for herd immunity) a false sense of accurracy.

Reviewer #2: The concluisions are appropriate

Reviewer #3: Line 315- Fragmenting large populations would need a better understanding of dog movements – these will vary between populations (effectively then altering the in-out migration by an unknown amount)

Line 319 – Busy roads can also facilitate dog movement, reinforcing the need for localised knowledge of dog movements before implementing a campaign to fragment the population

Line 354- rubbish dumps. This is really important and rarely reported evidence of the need for the local authorities/environmental department to be involved in rabies control

Line 356- excluding sub populations with low contact rates could be counterproductive, as although they may contribute less to overall maintenance, they may still be likely to bite humans and therefore cause disease

**Editorial and Data Presentation Modifications?**

Reviewer #1: Line 17 & 20: ‘rife’ & ‘onus’– as many of the readers are no English-native speakers, I would suggest using more common used words like ‘widespread’ and ‘responsibility’

Line 20: insert ‘human’; dog-mediated human rabies

Line 58-59: Rather strange statement; without high quality vaccines, you will never reach high herd immunity. See Bali as an example; low quality vaccines resulted in low herd immunity.

Line 63-64: more important is the extreme high birth rate in lowering the vaccination coverage, so instead of mentioning only the death rate here, it may be better to refer to the extreme high population turnover.

Line 71: the term ‘subpopulations’ may need some explanation at this stage as it can refer to completely different aspects. For example, are spatial separated populations, different age groups, or subpopulations based on their level of confinement meant. The authors consider only population size in this manuscript.

Line 77: insert ‘that’ – ‘… subpopulation that are relatively …’

Line 86: at the end of the sentence, correct ‘thes’ to ‘these’

Line 339-340: It is also easier for a rabid dog to contact other dogs in the population if all dogs are living in 1km² than if they are spread out over 100km². Ignoring (sub)population density completely is not acceptable, especially considering coming up with an explanatory model containing 76 terms.

Table 2: it may be better to change the currency to U$ or €.

Line 361-388: A very interesting example comparing the effect of vaccination versus sterilization and the associated costs

Line 411-413: I would suggest deleting this sentence and reference as it is confusing. Rabies vaccines for dogs of good (standard) quality are protected against infection. Claiming that rabies in vaccinated dogs causes predominantly paralytic rabies is therefore misleading.

Reviewer #2: Please see summary of comments

Reviewer #3: Line 68 ‘vaccine’ campaigns is more accurate than ‘campaigning’

Line 22- Define ‘success’- (e.g elimination, reduction in canine cases, reduction in human cases)

 Line 26- ‘some additional factors’- would help the reader to be more specific here

Line 32- ‘lower fecundity’ appears to contradict line 23

Line 55. Stating it is a rare disease is in contrast to the later statements about being globally endemic

Line 58 – counterfeit or low quality vaccines have been a limiting factor in several outbreaks

Line 76 missing word(s) This study attempts to address these needs so that campaign managers can target sub-populations are relatively susceptible to elimination.

Line 98 thes

Line 86 ‘ due to the stochastic nature of the model’. Suggest changing the order so the model is introduced first.

Also need to define ‘successes’ and ‘failures’

Line 104- Anderson 2019. not formatted like other references

Line 142. A reference would help support this suggestion

Line 160- do you mean simulations?

Line 236 ‘betweeness centrality’ needs some explanation

Line 342 ‘with double of half the contact rates’ 

Line 377 and 378- are the costs the wrong way round? They are different in the table- vaccination costs 9134.01 

Table 2- these are results and would be better in the results rather than the discussion

Line 409. May be more accurate to say ‘in itself can be an awareness campaign’. In many regions the awareness is sufficient prior to the campaigns, but there are other reasons for not engaging 

Line 411 rabies signs not ‘symptoms’

**Summary and General Comments**

Reviewer #1: A very interesting approach, looking at subpopulations of dogs and the probability of elimination. This is an increasing important topic; how to use the limited resources (funds, vaccines, personnel, etc.) in the most cost-effective way. Here, only the effect of population size was considered. Looking at the level of restriction would be another way: the free-roaming dog population is key in the transmission of rabies. So, 60% overall vaccination coverage incl. well-confined owned dogs is not the same as 60% vaccination coverage in the free-roaming dog population. The major outcome of the study is unfortunately an open door “elimination of rabies in smaller populations is more likely than in larger population”, just as “elimination of animals as a result of rabies is more likely in smaller populations than in larger populations” (see Ethiopian wolves and African hunting dogs). One does not need a regression model to predict this. And here is my major issue with this manuscript. The complexity of rabies transmission is reduced to a simplified stochastic model (simple frequency dependent transmission instead of the more complex mixture of frequency – and density-dependent transmission - see specific comment line 112-125), subsequently an extremely complicated regression model with 76 (!!!) terms, incl. quadratic and cubic (interactions) terms is selected. A much simpler model that can easily be checked by anyone would have given the same validity, as the input values are based on the outcome of a disease transmission model with questionable assumptions. Hence, to interpret the outcome of a very (read: too) complicated regression model based on a simplified disease transmission model is in my opinion not valid.

Reviewer #2: The paper by Kotzé et al. entitled Modelling the factors affecting the probability for local rabies elimination by strategic control investigates the effects that six factors have on the probability of elimination of canine-mediated rabies. Given the current consensus to eliminate human rabies by 2030, this study is of interest. 

The paper is well written in good English and, although I do not have a good prior knowledge of the stochastic model (Anderson, 2019) used in this paper, the analyses seem sound. But someone with better knowledge than me would be needed to confirm the model’s suitability and appropriate use. 

My principle comment is with regards to inappropriate placement of text in Methods, Results and Discussion which makes comprehension tricky. I will give examples in the list of comments below.

Line:

51: Introduction – needs to be some discussion of disease modelling, especially of the type used in the paper.

52: dogs don’t transmit human deaths

53: are we sure scratching is an exposure?

55: what does relative rarity mean? Need to define this a bit.

66: ‘vary drastically’ - suggest couch in less strident terms – eg. ..will likely vary between locations…

77: missing word – sub-populations are relatively ..

86: thes 

88: numbers need commas, eg 237,224

126 - 129: there are papers where unvaccinated dogs which are sero-positive have been reported. Eg. http://dx.doi.org/10.1016/j.vaccine.2016.10.015 found 1.3% unvaccinated dogs to be sero-pos. This suggests that the ‘exposure survival’ range could include a higher maximum level than 0.001. 

160: It may have been explained earlier, but I think there is a need to (re-)explain why five iterations were computed. Also this sentence is confusing and suggest re phrase for clarity.

162-3: Simple scatter plotes …..with elimination – This should be in Results.

185-6: The predicted ….Figs 1-5. This should be in Results.

All Figs should be listed in the Results

This should be in Results.

205: Boxplot text - This should be in Results.

235: Fig 11 - This should be in Results.

271: Fecundity and In-Migration section – Fig 4 showing the In-Migration results indicate very little if any effect of In-Migration on p(elim). This is not really explained in the results, which, whilst describing a very marginal effect, give the impression that there is an impact (line 276 – 278). In addition, the best fit line of the 30% Immune plot from Fig 4 doesn’t seem to fit the dots very well. The dots suggest that there should be more of a curve. 

280-2: Perhaps this text should be in Discussion

287: give which sub-plot this is in. Again, as stated in line 285, the effect seen in these plots is overwhelmingly as a result of the % of immune dogs; the description of the results in line 287-8 could make this more explicit. Eg. For example, reducing the probability of surviving viral exposiure from 1% to zero only increased p(elim) by a very small amount (0.58 to 0.6).

298: Need to discuss fig 11 in Results

303: not ‘rate’: higher probability of elimination.

312-13: Need to describe this in Results

326: might be good to give some brief explanation of what ‘betweenness centrality’ means 

346: Understanding …..is crtical to understand…. Rephrase

348-350: this analysis should be described in Methods and Results

350: ‘we find that the elimination probability increases from 0.6 to 0.99 as the infectivity drops from 1 to 0.6…’ – isn’t this obvious given that ‘infectivity’ is a proxy for R0?

350-1: we find that ….(equal to a contact rate…..in NDamena example) – its not clear how this is equal

369: The Table 2 example should be in Methods and Results

377 onwards: suggest all cost values given in USD

Table 2: 

Row 3: 201.66 bitches = 202 bitches

Row 4: from 40% to 44% is (this is not appropriate text for a cell)

Row 4: from 0 to 10% is 10% (suggest re-phrase)

Row 7: Bracket of pop captured – need explanation for these ranges.

392: ‘realise,’ doesn’t need a comma

402: there is evidence in literature for seroconversion without lethal infection in dogs

407: more so since …no need for comma

Reviewer #3: This is a well written and thorough study on an important zoonosis. 

The work is an advance on previous work- showing the sensitivity of a previously published model on rabies to various factors, but the specific areas that are covered in this manuscript will become increasingly important as we move toward elimination of human mediated dog rabies .

One major limitation the authors acknowledge is that it is not spatially explicit, which has impilcations for the inferences and application to different areas, but does not detract from the conclusions that the authors draw from the data.

Another important potential limitation is that the model doesn't include the introduction of rabid dogs into the population. This is a very likely scenario in many endemic areas, and therefore including a discussion about where that assumption would hold would enhance the paper (e.g islands)

PLOS authors have the option to publish the peer review history of their article (what does this mean?). If published, this will include your full peer review and any attached files.

Reviewer #1: No

Reviewer #2: No

Reviewer #3: No
---

## [Decision Letter · Decision Letter 1]

3 Feb 2021

Dear Dr Kotze,

Thank you very much for submitting your manuscript "Modelling the factors affecting the probability for local rabies elimination by strategic control" for consideration at PLOS Neglected Tropical Diseases. As with all papers reviewed by the journal, your manuscript was reviewed by members of the editorial board and by several independent reviewers. The reviewers appreciated the attention to an important topic. Based on the reviews, we are likely to accept this manuscript for publication, providing that you modify the manuscript according to the review recommendations. 

Please review and address the outstanding comments from reviewers in preparing your revision, particularly the discussion regarding the time frame of simulations as it relates to strategic planning for multi-year canine rabies control programs by managers. 

Sincerely,

Amy T. Gilbert

Deputy Editor

Sergio Recuenco

Deputy Editor

Reviewer's Responses to Questions

**Key Review Criteria Required for Acceptance?**

**Methods**

-Are the objectives of the study clearly articulated with a clear testable hypothesis stated?

-Is the study design appropriate to address the stated objectives?

-Is the population clearly described and appropriate for the hypothesis being tested?

-Is the sample size sufficient to ensure adequate power to address the hypothesis being tested?

-Were correct statistical analysis used to support conclusions?

-Are there concerns about ethical or regulatory requirements being met?

Reviewer #1: The proposed model is clearly described and also the assumptions made are explained in detail. Sometimes, I got the feeling that the model did not have to 'fit' the real world observations but the other way around; e.g. the number of secondary contacts from a rabid dog follows a negative binomial distribution with mean X and variance Y. I always thought it was the other way around a negative binomial distribution with mean X and variance Y approaches the number of secondary contacts from a rabid dog.

Reviewer #2: THE METHODS WERE CLEAR. ONLY ONE COMMENT BELOW.

line 192-4: We discourage campaign managers to plan resource allocations based on probabilities of elimination beyond one year following a campaign and therefore stopped simulations after one year. THIS IS CONFUSING AND REQUIRES BRIEF CLARIFICATION.

Reviewer #3: (No Response)

**Results**

-Does the analysis presented match the analysis plan?

-Are the results clearly and completely presented?

-Are the figures (Tables, Images) of sufficient quality for clarity?

Reviewer #1: Also, results are presented in detail and the figures can easily be interpreted by the reader

Reviewer #2: RESULTS ARE CLEAR; APART FROM ONE POINT:

Line 410-12: its not clear why the procedural costs were low and capture costs high - this was partly becasue Table 1 was too wide and was not completely visible in the pdf.

Reviewer #3: (No Response)

**Conclusions**

-Are the conclusions supported by the data presented?

-Are the limitations of analysis clearly described?

-Do the authors discuss how these data can be helpful to advance our understanding of the topic under study?

-Is public health relevance addressed?

Reviewer #1: The conclusions and recommendations are supported by the data presented. However, most conclusions and recommendations are already well established and are outlined in many dog rabies management plans. The only new concept in the immune corridor. However, the effectiveness of these corridors is not supported by data but merely speculative. The authors have clearly outlined the public health relevance of their findings

Reviewer #2: ALL CLEAR; ONE POINT:

LIne 533 - missing word.... 'the' surveillance system.

Reviewer #3: (No Response)

**Editorial and Data Presentation Modifications?**

Reviewer #1: Line 58-61: I think the low incidence rate (0.2%) is not a result of the low R-rate (<1.5). Look at COVID here R is also below 1.5 but we see in some countries high number of infections. The low incidence rate in endemic countries is because not all dogs infected (bitten) will develop rabies and in almost all endemic countries a certain percentage of the dog population is vaccinated.

Line 100-101: Interesting to see that density is not selected. In line 138-147, the rationale for not selecting density-dependent transmission is satisfactorily explained in detail. However, to claim that the latter was suggested as an alternative for frequency-dependent transmission is historically not correct. Not long ago, rabies was considered an exclusively density-dependent transmitted disease. Fortunately, this has changed in recent years and also frequency-dependent transmission has been suggested. 

Line 192-194: As a non-native speaker I may have misinterpreted this; if you have not eliminated rabies within one year after a campaign, don’t even bother? I hope, I misinterpreted this but then it should be rewritten to circumvent misunderstandings. Rabies elimination can take years in order to break the transmission cycle. So, this is a disturbing piece of advise.

Line 405: Maybe I misinterpret it, but there are no capture costs for vaccination only for sterilization? In areas with a high number of free-roaming dogs, a certain percentage of animals cannot be easily restrained and thus to achieve high vaccination coverages in these areas would include CVR-techniques associated with high capture costs.

Line 450-454: I agree with the authors. However, some outbreaks in the field show a different scenario, like Flores and Bali. Two islands where a single dog introduced a rabies outbreak in 1998 (Flores) and 2008 (Bali) that are still not brought under control. If surveillance and awareness is not well established, rabies can become widespread (temporal and spatial) before it is detected (beyond secondary contacts). 

Line 525-528: This sentence could use a reference like: Jean S et al (2019) Determining the post-elimination level of vaccination needed to prevent re-establishment of dog rabies. PLoS Negl Trop Dis. 13(12):e0007869.

Reviewer #2: Accept

Reviewer #3: (No Response)

**Summary and General Comments**

Reviewer #1: Certain assumptions made on behavioural ecology and disease transmission are oversimplifications and could have a significant impact on the outcome of the model. For example, the statement that ‘reduced fecundity equates to reduced population turnover rates’ (line 149-150). Population turnover is affected by reproduction, death, dispersal (emigration and immigration) and the interaction between these factors. So, reduced fecundity does not always have to lead to lowered turnover rates, esp. in disease ecology. However, the outcome of the model pretty much confirms what is already known. Most conclusions and recommendations are also not new; movement restrictions, interrupt access to garbage dumps, etc. An interesting new concept is ‘immune corridor’. However, the added value of this is not supported by data from the model but an idea to reach the goals of the model. Although interesting, it remains to be proven if it would work. It may be more appropriate for wildlife rabies than dog rabies as dog translocations by humans always jeopardize such corridors.

Reviewer #2: This is the second time I have reviewed the paper. This resubmission is much improved and the paper is now very well written, with clear conclusions drawn. Apart from the minor comments given above, I have no further comments to make.

Reviewer #3: The authors have thoroughly addressed most of the comments, and the manuscript is much improved

One remaining issue that may need further clarification is the reason given for stopping simulations after one year. The authors state that they want to use the results of this study to 'allocate resources strategically' (line 94) and yet the methods used appear to be driven by, and restricted by, the current practice- ' We discourage campaign managers to plan resource allocations based on probabilities of elimination beyond one year following a campaign and therefore stopped simulations' (line 191). This means that the study will never be able to provide longer term strategic guidance in its current form. The study would seem to be much stronger and potentially more useful if the simulations could be run for more than one year (and include more than one campaign). 

If it is not possible to extend the model simulations for technical reasons, then I would suggest the study would benefit from it being made clearer in the title and throughout, that the study is modelling the impact of a single vaccination campaign, and in the limitations (line 551) be more explicit about the probability (from previous studies with BioEcon) that elimination might occur in year 2 , with a reference .

PLOS authors have the option to publish the peer review history of their article (what does this mean?). If published, this will include your full peer review and any attached files.

Reviewer #1: No

Reviewer #2: No

Reviewer #3: No
---

## [Editor Report · Decision Letter 2]

11 Feb 2021

Dear Dr Kotze,

We are pleased to inform you that your manuscript 'Modelling the factors affecting the probability for local rabies elimination by strategic control' has been provisionally accepted for publication in PLOS Neglected Tropical Diseases.

Best regards,

Amy T. Gilbert

Deputy Editor

Sergio Recuenco

Deputy Editor

I thank the authors for providing access to all of the data and analysis workflow. Furthermore, the authors have adequately addressed the most recent set of concerns raised by reviewers.

---

## [Editor Report · Acceptance letter]

28 Feb 2021

Dear Dr Kotzé,

We are delighted to inform you that your manuscript, "Modelling the factors affecting the probability for local rabies elimination by strategic control," has been formally accepted for publication in PLOS Neglected Tropical Diseases.

Best regards,

Shaden Kamhawi

co-Editor-in-Chief

Paul Brindley

co-Editor-in-Chief
